# Integrated cross-study datasets of genetic dependencies in cancer

Clare Pacini [1,2], Joshua M. Dempster [3], Isabella Boyle [3], Emanuel Gonçalves [1], Hanna Najgebauer[1,2,4], Emre Karakoc[1,2], Dieudonne van der Meer[1], Andrew Barthorpe[1], Howard Lightfoot[1], Patricia Jaaks[1], James M. McFarland [3], Mathew J. Garnett [1,2], Aviad Tsherniak [3] & Francesco Iorio [1,2,5✉]

CRISPR-Cas9 viability screens are increasingly performed at a genome-wide scale across large panels of cell lines to identify new therapeutic targets for precision cancer therapy. Integrating the datasets resulting from these studies is necessary to adequately represent the heterogeneity of human cancers and to assemble a comprehensive map of cancer genetic vulnerabilities. Here, we integrated the two largest public independent CRISPR-Cas9 screens performed to date (at the Broad and Sanger institutes) by assessing, comparing, and selecting methods for correcting biases due to heterogeneous single-guide RNA efficiency, gene-independent responses to CRISPR-Cas9 targeting originated from copy number alterations, and experimental batch effects. Our integrated datasets recapitulate findings from the individual datasets, provide greater statistical power to cancer- and subtype-specific analyses, unveil additional biomarkers of gene dependency, and improve the detection of common essential genes. We provide the largest integrated resources of CRISPR-Cas9 screens to date and the basis for harmonizing existing and future functional genetics datasets.

[1] Wellcome Sanger Institute, Wellcome Genome Campus, Hinxton, Cambridge, UK. [2] Open Targets, Wellcome Genome Campus, Hinxton, Cambridge, UK. [3] Broad Institute of MIT and Harvard, Cambridge, MA, USA. [4] European Molecular Biology Laboratory, European Bioinformatics Institute, Wellcome Genome Campus, Cambridge, UK. [5] Human Technopole, Milano, Italy. ✉email: francesco.iorio@fht.org

Cancer is a complex disease that can arise from multiple different genetic alterations. The alternative mechanisms by which cancer can evolve result in considerable heterogeneity between patients, with the vast majority of them not benefiting from approved targeted therapies[1]. In order to identify and prioritize new potential therapeutic targets for precision cancer therapy, analyses of cancer vulnerabilities are increasingly performed at a genome-wide scale and across large panels of in vitro cancer models[2–11]. This has been facilitated by recent advances in genome editing technologies allowing unprecedented precision and scale via CRISPR-Cas9 screens. Of particular note are two large pan-cancer CRISPR-Cas9 screens that have been independently performed by the Broad and Sanger institutes[2,12]. The two institutes have also joined forces with the aim of assembling a joint comprehensive map of all the intracellular genetic dependencies and vulnerabilities of cancer: the Cancer Dependency Map (DepMap)[13,14].

The two generated datasets collectively contain data from over 1000 screens of more than 900 cell lines. However, it has been estimated that the analysis of thousands of cancer models will be required to detect cancer dependencies across all cancer types[3]. Consequently, the integration of these two datasets will be key for the DepMap and other projects aiming at systematically probing cancer dependencies. These integrated datasets will provide a more comprehensive representation of heterogeneous cancer types and form the basis for the development of effective new therapies with associated biomarkers for patient stratification[15]. Further, designing robust standards and computational protocols for the integration of these types of datasets will mean that future releases of data from CRISPR-Cas9 screens can be integrated and analyzed together, paving the way to even larger cancer dependency resources.

We have previously shown that the pan-cancer CRISPR-Cas9 datasets independently generated at the Broad and Sanger institutes are consistent on the domain of 147 commonly screened cell lines[16]. The reproducibility of these CRISPR screens holds despite extensive differences in the experimental pipelines underlying the two datasets, including distinct CRISPR-Cas9 sgRNA libraries. Here, we investigate the integrability of the full Broad/Sanger gene-dependency datasets, yielding the most comprehensive cancer dependency resource to date, encompassing dependency profiles of 17,486 genes across 908 different cell lines that span 26 tissues and 42 different cancer types. We compare different state-of-the-art data-processing methods to account for heterogeneous single-guide RNA (sgRNA) on-target efficiency, and to correct for gene independent responses to CRISPR-Cas9 targeting[12,17,18], evaluating their performance on common use cases for CRISPR-Cas9 screens (Fig. 1a, b, and c).

We show that our integration strategy accounts and corrects for technical biases while preserving gene-dependency heterogeneity and recapitulates established associations between molecular features and gene dependencies. We highlight the benefits of the integrated dataset over the two individual ones in terms of improved coverage of the genomic heterogeneity across different cancer types, identification of new biomarker/ dependency associations, and increased reliability of human core-fitness/common-essential genes (Fig. 1d). Finally, we estimate the minimal size (in terms of the number of screened cell lines) required in order to effectively correct batch effects when integrating a new dataset.

Collectively, this study presents a robustly benchmarked framework to integrate independently generated CRISPR-Cas9 datasets that provide the most comprehensive resource for the exploration of cancer dependencies and the identification of new oncology therapeutic targets.

## Results

**Overview of the integrated CRISPR-Cas9 screens.** The Sanger's Project Score CRISPR-Cas9 dataset (part of the Sanger DepMap)[19] and the Broad's 20Q2 DepMap dataset[20,21] contain data for 317 and 759 cell lines, respectively. Overall, these represent screens for 908 unique cell lines (Fig. 2a and Supplementary Data 1). Together these cell lines spanned 26 different tissues (Fig. 2b) and for 16 of these, the number of cell lines covered increased when considering both datasets together. Similarly, the integrated dataset provided richer coverage of specific cancer types and clinically relevant subtypes (Fig. 2c). These preliminary observations highlight the first benefit of combining these resources to increase statistical power for tissue-specific, as well as pooled pan-cancer analyses.

Between the two datasets, there was an overlap of 168 cell lines screened by both institutes, encompassing 16 different tissue types (median = 8, min 1 for soft tissue, biliary tract and kidney, max 28 for lung, Fig. 2a, b). The set of overlapping cell lines enabled the estimation of batch effects due to differences in the experimental protocols underlying the two datasets[16], without biasing the correction toward specific cell line lineages.

**Data pre-processing.** Known biases in CRISPR screens arise due to nonspecific cutting toxicity that increases with copy number amplifications (CNAs)[22,23] and heterogeneous levels of on-target efficiency across sgRNAs targeting the same gene[24]. Multiple methods exist to correct for these biases. Here, we evaluate three: CRISPRcleanR, an unsupervised nonparametric CNA effect correction method for individual genome-wide screens[17]; a method resulting from using CRISPRcleanR with JACKS, a Bayesian method accounting for differences in guide on target efficacy[18] (CCR-JACKS) through joint analysis of multiple screens; and CERES, a method that simultaneously corrects for CNA effects and accounts for differences in guide efficacy[12], also analyzing screens jointly.

**Batch effect correction.** Technical differences in screening protocols, reagents and experimental settings can cause batch effects between datasets. These batch effects can arise from factors that vary within institute screens (for example, differences in control batches and Cas9 activity levels) as well as between institutes (such as differences in assay lengths and employed sgRNA libraries). When focusing on the set of cell lines screened at both institutes, a principal component analysis (PCA) of the cell line dependency profiles across genes (DPGs) highlighted a clear batch effect determined by the origin of the screen, irrespective of the pre-processing method, consistent with previous results (Fig. 3a)[16].

We quantile normalized each cell line DPG and adjusted for differences in screen quality in the individual Broad/Sanger datasets. The combined Broad/Sanger dataset was then batch-corrected using ComBat[25] (Methods). Following ComBat correction, the combined datasets on the overlapping cell lines showed reduced yet persistent residual batch effects clearly visible along the two first principal components (Supplementary Fig. 1). Analysis of the first two principal components (using MsigDB gene signatures[26] and all cell lines, Methods), showed enrichment for metabolic processes (phosphorus metabolic process $q$-value = 1.06e-08, protein metabolic process $q$-value = 8.70e-07, hypergeometric test) in the first principal component. The enrichment of metabolic processes is consistent with differences identified across these datasets due to different media conditions employed in the underlying experimental pipelines[16,27]. The second principal component contained significant enrichments for protein complex organization and assembly ($q$-value = 1.57e-16

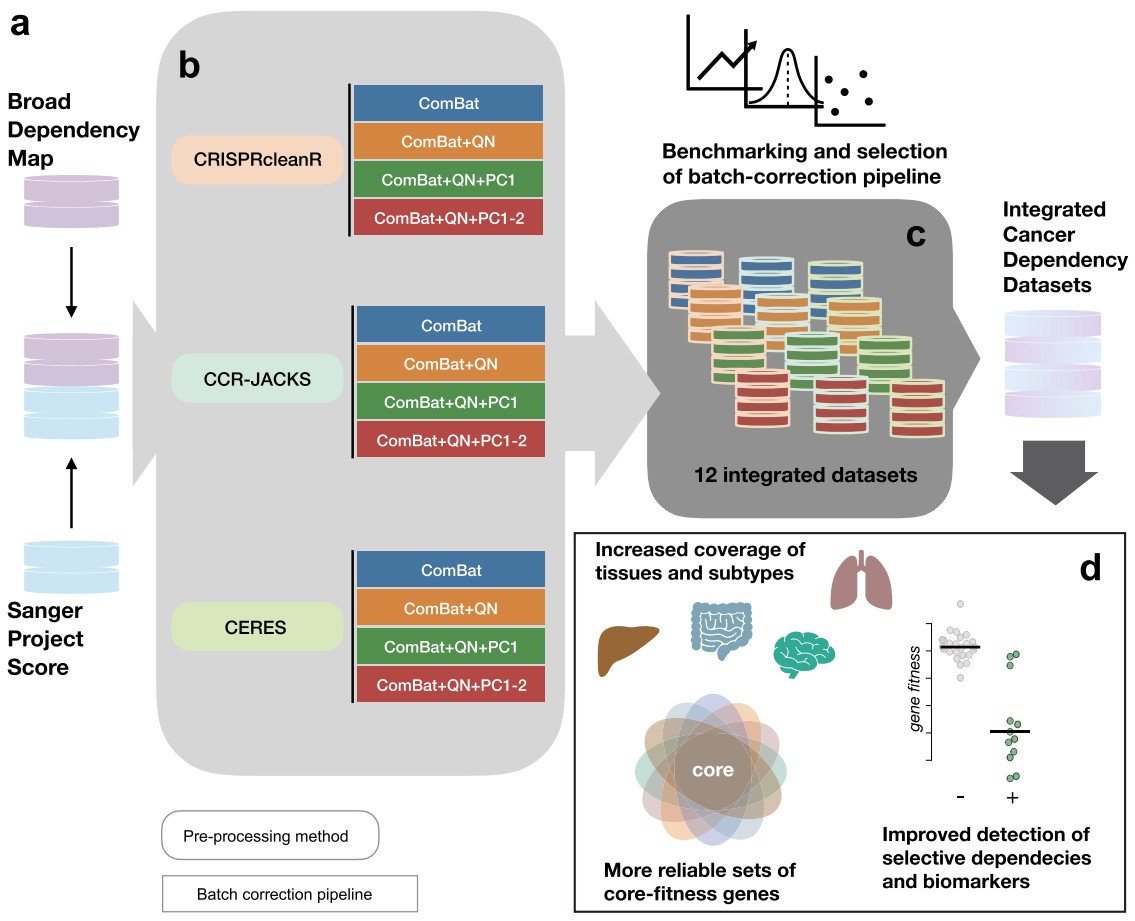

**Fig. 1 Schematic of the integration strategy. a** Broad and Sanger gene-dependency datasets (raw count data of single-guide RNAs) are downloaded from respective web-portals. **b** The datasets from each institute are pre-processed with three different methods, accounting for gene-independent responses to CRISPR-cas9 targeting (arising from copy number amplifications) and heterogeneous sgRNA efficiency, providing gene-level corrected depletion fold changes. Then, four different batch-correction pipelines are applied to the gene-level fold changes across the two institute datasets for each of the pre-processing methods. **c** Twelve different integrated datasets resulting from applying three different pre-processing methods CRISPRcleanR, CRISPRcleanR with JACKS (CCR-JACKS) and CERES (as indicated by the border colors) and four different batch-correction pipelines ComBat only (ComBat), ComBat with quantile normalization (ComBat + QN) and ComBat + QN with either the first principal component (PC) removed (ComBat+QN + PC1) or first two PCs removed (ComBat+QN + PC1-2) (as indicated by the fill colors) are benchmarked. **d** Advantages provided by the final integrated datasets and conservation of analytical outcomes from the individual ones are investigated.

and 5.28e-11, respectively, hypergeometric test) (Supplementary Data 2), which have no obvious associations with technical biases found in CRISPR-cas9 screens. Based on these results, we considered four different batch correction pipelines and evaluated their use in our integrative strategy. In the first pipeline, we processed the combined Broad/Sanger DPG dataset using ComBat alone (ComBat). In the second, we applied a second round of quantile normalization following ComBat correction (ComBat + QN) to account for different phenotype intensities across experiments, resulting in different ranges of gene-dependency effects. In the third and fourth pipelines we also removed the first one or two principal components, respectively, (ComBat+QN + PC1) and (ComBat+QN + PC1-2).

The final 12 datasets contained data from unique screens of 908 cell lines using each of the three pre-processing methods and four different batch correction pipelines as outlined in the previous section. To assess the performance of different batch correction pipelines we estimated, using the overlapping cell lines, the extent to which each cell line DPG from one study matched that of its counterpart (derived from the same cell line) from the other study following batch correction. To quantify the agreement, we calculated for each DPG its similarity to all other screen DPGs using a weighted Pearson's (wPearson) correlation (Methods). We then calculated the proximity of a cell line to its counterpart compared to all other cell lines using the wPearson as a metric (recall of cell line identity) (Fig. 3b).

The best performances were obtained when removing either the first or the first two principal components following ComBat and quantile normalization, i.e., ComBat+QN + PC1 or ComBat + QN + PC1-2. Across pre-processing methods, CERES performed best with 302 (90%) of the cell lines being closest to their counterpart from the other study ($k = 1$) followed by CRISPRcleanR with 272 cell lines (81%) and CCR-JACKS with 215 (64%). The recall of cell line identity was high for each integration pipeline with normalized Area under the curve (nAUC) values of 0.98 for CCR-JACKS and 0.99 for CRISPRcleanR and CERES when considering the best performing ComBat+QN + PC1-2 batch correction method.

**Performance of the integration pipelines.** We evaluated the performance of each of the 12 integrated datasets, containing 908 cell lines, under four use-cases: the identification of (i) essential and non-essential genes (ii) lineage subtypes (iii) biomarkers of selective dependencies, and (iv) functional relationships.

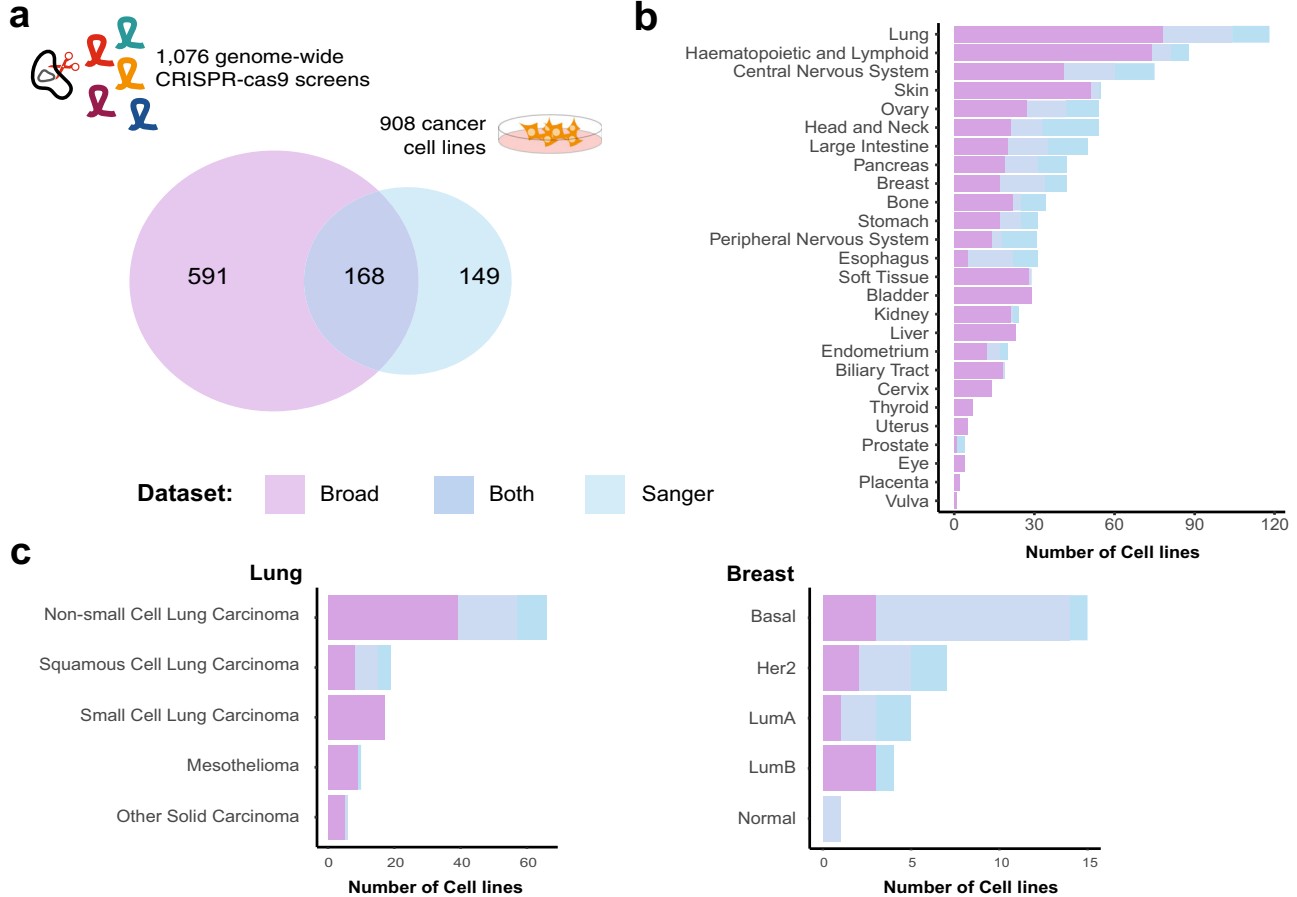

**Fig. 2 Overview of CRISPR-Cas9 screened cancer cell lines. a** Number of cell lines screened by the Broad and the Sanger institutes and their overlap. **b** Overview of the number of cell lines screened for each tissue type across the two datasets. **c** Number of screened lung cancer and breast cancer cell lines split according to cancer types and clinical PAM50 subtypes, respectively, across the two datasets.

**Identification of essential and non-essential genes.** A cell line DPG with a large separation of dependency scores (DS) of common essential and non-essential genes should yield lower misclassification rates when identifying dependencies specific to that cell line. For each cell line we measured the separation of dependency scores (DS) between known common essential and non-essential genes[11] across all integrated datasets. As a measure of separation we used the null-normalized mean difference (NNMD)[28], defined as the difference between the mean DS of the common essential genes and non-essential genes divided by the standard deviation of the DSs of the non-essential genes.

By analyzing multiple screens jointly, CERES and JACKS borrow essentiality signal information across screens. As a consequence, these methods better identify consistent signals across cell line DPGs (i.e., for common essential and non-essential genes), especially for DPGs derived from lower quality experiments, or reporting weaker depletion phenotypes[18,23]. Consistently, CERES (median NNMD range $[-5.78, -5.88]$) showed better NNMD values than CRISPRcleanR (median NNMD range $[-5.02, -5.12]$, Wilcox test (WT) $N = 908$ largest $p$-value 1.86e-114) and CCR-JACKS (median NNMD range $[-5.14, -5.23]$, $N = 908$ WT largest $p$-value 3.58e-111)), and similarly CCR-JACKS had better NNMD values than CRISPR-cleanR (largest WT $p$-value 0.00021, $N = 908$) (Fig. 4a). Comparing the batch correction methods, ComBat+QN + PC1-2 had marginally better performance across all pre-processing methods.

Next, we evaluated the gene-dependency false-positive rates across all integrated datasets. For each cell line DPG, we defined a set of putative negative controls composed of genes not expressed at the basal level in that cell line (Methods). False positives were calculated as the sum of negative controls identified as significant dependencies (in the top 15% most depleted genes) normalized by their total number across the DPG. There was little difference in false-positive rates across the four different batch correction pipelines, with a slight improvement when two principal components were removed (Fig. 4b). CERES outperformed CCR-JACKS significantly for all batch correction methods (largest $\chi^2$ contingency table $p$-value $1.87 \times 10^{-11}$, $N = 1.43 \times 10^7$) and CCR-JACKS outperformed CRISPRCleanR ($p$-value below machine precision). Comparing the correction methods, the differences between ComBat and ComBat+QN and between ComBat+QN + PC1 and ComBat+QN + PC1-2 were generally not significant across pre-processing methods, while the difference between either ComBat or Combat+QN and either ComBat +QN + PC1 or ComBat+QN + PC1-2 were generally significant (largest $p$-value $1.42 \times 10^{-5}$). As a final test of control separation, we used the unexpressed genes as an empirical null distribution for each DPG to estimate $p$-values for all DS and thus false-discovery rates (FDRs) within each DPG. We calculated the recall of a reference set of common essential genes[11] at 10% FDR (Fig. 4c). Again, CERES outperformed CCR-JACKS, which outperformed CRISPRCleanR, and increasing the number of steps in the batch correction pipeline monotonically improved essential recall for all pre-processing methods. All differences between pre-processing methods and batch correction methods were significant, with the largest observed $t$-test (related) $p$-value $1.96 \times 10^{-3}$ ($N = 830$).

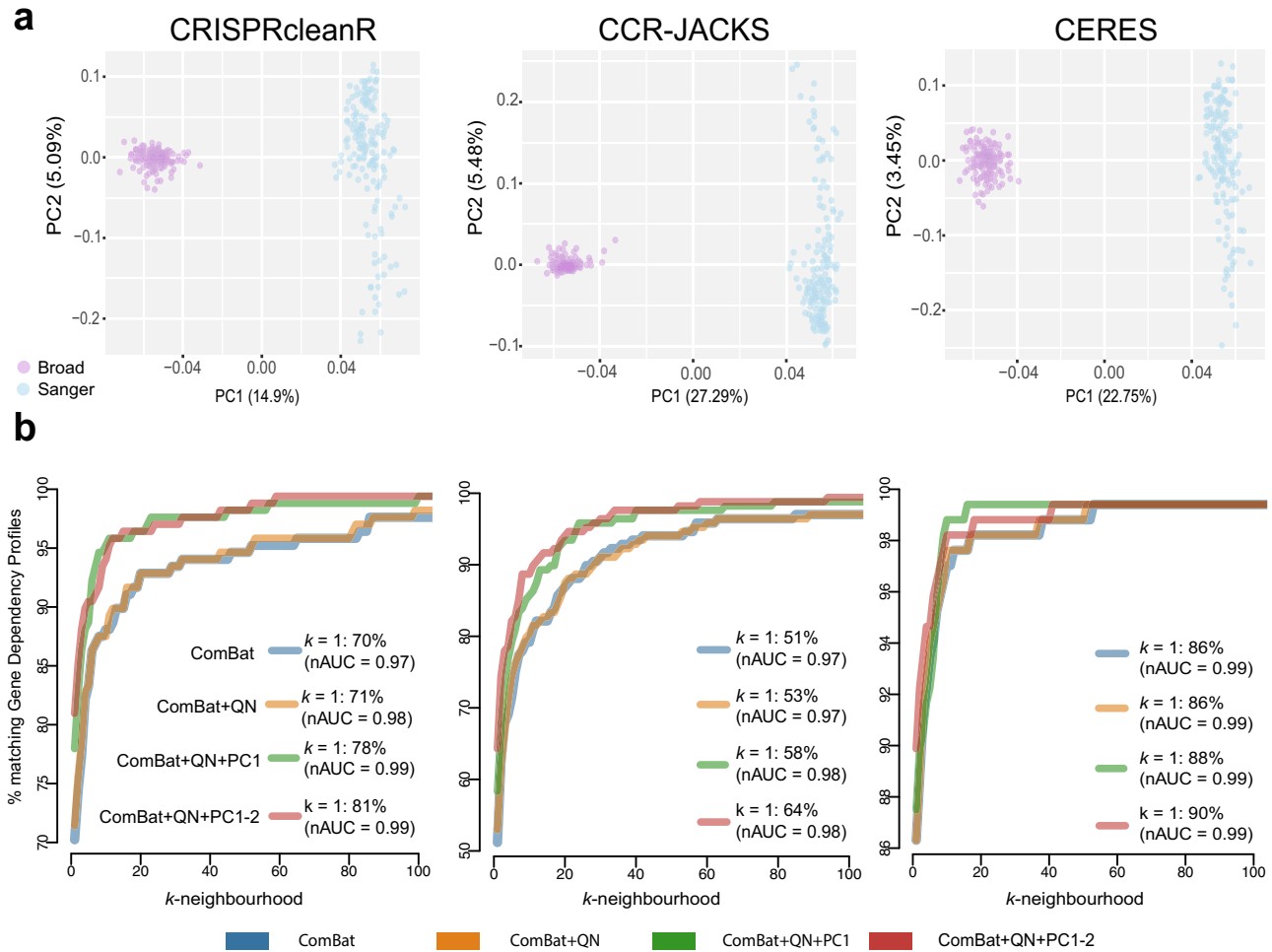

**Fig. 3 Batch effect assessment and correction. a** Principal component plots of the dependency profile across genes (DPGs) for cell lines screened in both Broad and Sanger studies and pre-processing methods. Screens are colored by the institute of origin with purple indicating a screen performed at the Broad and blue a screen from the Sanger. **b** Percentages of cell line DPGs that have the corresponding (same cell line) DPG screened at the other institute among their $k$ most correlated DPGs (the $k$-neighborhood). Results are shown across different pre-processing methods, CRISPRcleanR, CRISPRcleanR with JACKS (CCR-JACKS) and CERES (in different plots) and different batch correction pipelines (as indicated by the different colors). The different batch correction pipelines are ComBat only (ComBat, blue), ComBat followed by quantile normalization (ComBat + QN, orange), and ComBat + QN with either the first PC removed (ComBat+QN + PC1, green) or first two PCs removed (ComBat + QN + PC1-2, red). Correlations between DPGs are computed using a weighted Pearson correlation metric. Genes with higher selectivity have a larger weight in the correlation calculation. As a measure of selectivity, we used the average (across the two individual datasets) skewness of a gene's dependency profile across cell lines. The proportion of cell lines closest to their counterpart from the other study ($k = 1$) is shown and the normalized areas under the curves (nAUC) are shown in brackets. The x-axis values are restricted to between 1 and 100 to highlight the range over which performance differences are visible between datasets.

**Identification of lineage subtypes.** Many dependencies are context specific, reducing cellular fitness in a subset of lineages, that can be used to elucidate gene function and identify cancer type-specific vulnerabilities. To evaluate the ability of the integrated datasets in recapitulating tissue lineages and clinical subtypes we first estimated the extent of conserved similarity between screens of cell lines derived from the same tissue lineage. We evaluated the tendency of screens of cell lines from the same lineage to yield similar results by comparing unsupervised clustering of the batch-corrected cell line DPGs to the lineage labels of the cell lines. To this aim, we performed one hundred $k$-means clustering of each of the 12 datasets, with $k$ equal to the number of tissue lineages screened in at least one study. We then calculated the adjusted mutual information (AMI, Methods) between each DPG clustering and the partition of the cell lines induced by their lineage labels. We observed higher than chance AMI between the obtained $k$ clusters and the tissue lineages of the cell line DPGs, regardless of the starting batch-corrected dataset

(largest single-sample $t$-test $p$-value of $3.59 \times 10^{-135}$, $N = 100$, Fig. 4d). Under each pre-processing method the removal of one or two principal components resulted in an increased AMI between cell line DPGs clusters and tissue lineages.

We next measured the ability of each of the integrated datasets to separate cell lines according to lineage subtypes. The integrated datasets contain over 100 lung cell lines. These cell lines can further be stratified into subtypes such as small cell lung carcinoma and mesothelioma, while clinical subtypes such as PAM50 classifications are available for the breast cancer cell lines (Fig. 2c). To quantify the clustering of cell lines by subtype we calculated the correlation between all cell lines DPGs, and for a given query cell line the rank of the cell line with most correlated DPG to the query from the same subtype ($k$-rank). For the lung cancer cell lines, the percentage of cell lines whose closest neighbor was from the same subtype ($k = 1$) was greatest for CERES (64–65% across batch correction methods) followed by CRISPRcleanR (61–64%) and CCR-JACKS (50–57%), with slight

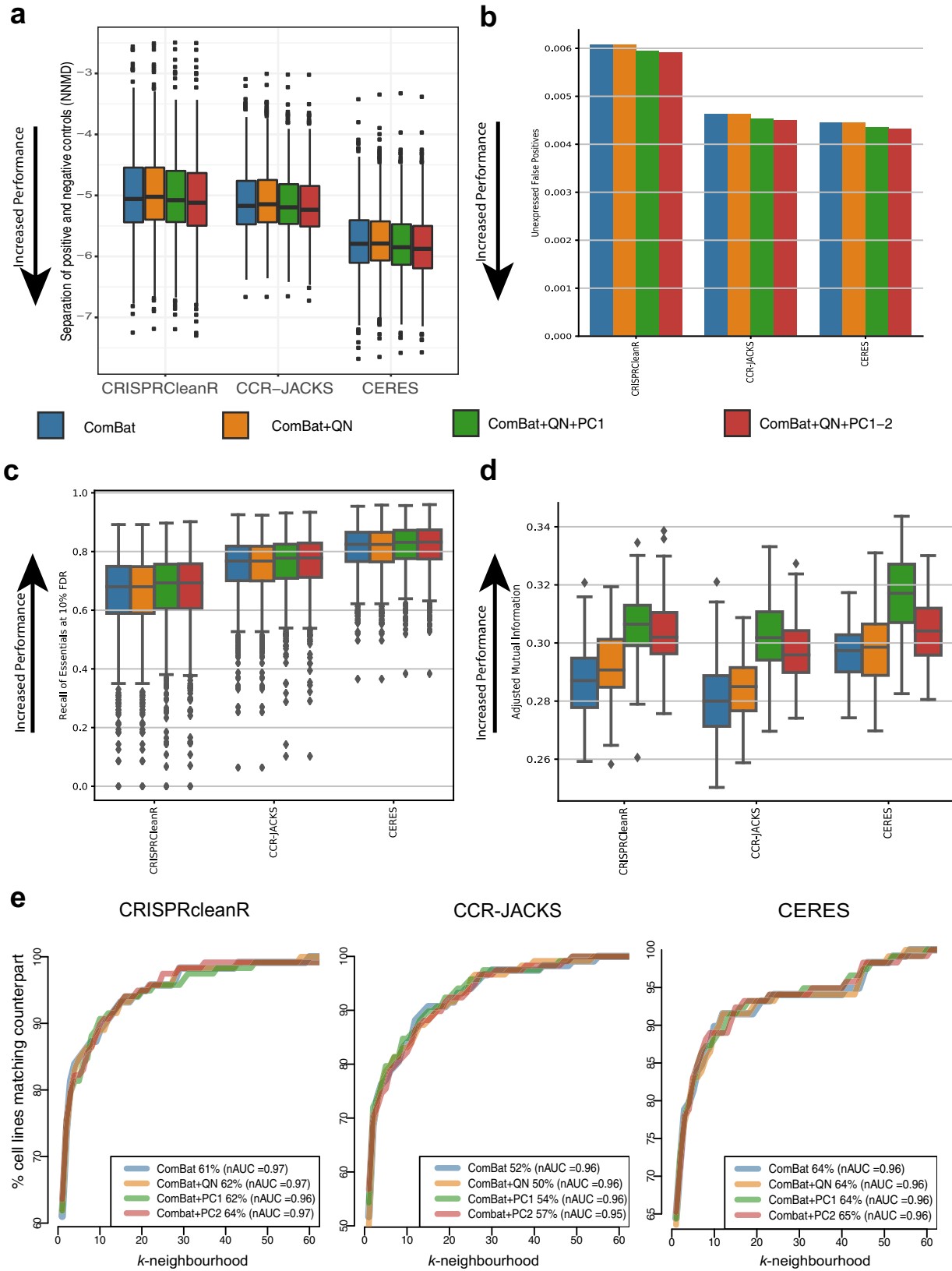

improvement with the removal of one or two principal components (Fig. 4e). The normalized area under the curve (nAUC) values showed little variation across batch correction methods and were broadly similar between the pre-processing methods CERES (lung = 0.96, breast = 0.91−0.92), CCR-JACKS

(lung = 0.95−0.96, breast = 0.84−0.85), CRISPRcleanR (lung = 0.96−0.97, breast = 0.89−0.9)(Supplementary Fig. 2).

**Identification of biomarkers**. Interesting potential novel therapeutic targets are genes that show a pattern of selective

**Fig. 4 Use case recall of essential genes and lineage identification. a** Null-normalized mean difference (NNMD, a measure of separation between dependency scores of prior-known essential and non-essentials genes): defined as the difference in means between dependency scores of essential and non-essential genes divided by standard deviation of dependency scores of the non-essential genes. More negative values of NNMD indicate better separation of essential genes and non-essential genes. The results are shown for the three different pre-processing methods, CRISPRcleanR, CRISPRcleanR with JACKS (CCR-JACKS) and CERES used with four different batch correction pipelines. The different batch correction pipelines were ComBat only (ComBat, blue), ComBat followed by quantile normalization (ComBat+QN, orange), and ComBat+QN with either the first PC removed (ComBat+QN + PC1, green) or first two PCs removed (ComBat+QN + PC1-2, red). Boxplots for $N = 908$ cell lines are drawn with the interquartile range within the boxes and the median as a horizontal line. The whiskers extend to 1.5 times the size of the interquartile range and points outside this range. **b** False-positive rates across all pre-processing methods and batch-correction pipelines. In the gene-dependency profile of a given cell line, a significant dependency gene was called a false positive if that gene was not expressed in that cell line. **c** Recall of known essential genes across all pre-processing methods and batch-correction-pipelines at 10% FDR using two-sided $t$-test for $N = 830$ cell lines. The boxplot shows the interquartile range in the box with the median as a horizontal line. Whiskers extend to 1.5 times the interquartile range and outlier points outside this range are plotted. **d** Agreement between cell line clusters based on the dependency profile across genes correlation and tissue lineage labels of corresponding cell lines, across pre-processing methods and batch-correction pipelines. Comparisons of datasets were performed using a two-sided $t$-test between $N = 100$ k-means clustering's. The box contains the interquartile range with the median as a horizontal line. Whiskers extend to 1.5 times the interquartile range and outlier points outside this range plotted. **e** Agreement of lung CRISPR-cas9 fitness profiles according to the lung cancer subtypes ($N = 118$). For each query lung cancer cell line in turn we computed Pearson correlation scores to all other lung cancer cell lines (responses). We then ranked the response cell lines according to these correlations. For each query cell line, the rank position $k$ of the most correlated response cell line from the same cancer subtype (matching response) was identified. A rank of $k = 1$ indicates that the query cell line was closest to another cell line from the same cancer subtype. The curves show the ratio of query cell lines with a matching response within a given rank position. The proportion of query cell lines with a matching response in $k = 1$ are also shown as percentages for each dataset. The normalized area under the curve (nAUC) for each dataset is shown in brackets. The figure shows the $x$-axis zoomed in to between 0 and 60.

dependency, i.e., exerting a strong reduction of viability upon CRISPR-Cas9 targeting in a subset of cell lines. Furthermore, these selective dependencies are often associated with molecular features that may explain their dependency profiles (biomarkers). We investigated each of the integrated datasets' ability to reveal tissue-specific biomarkers of dependencies. As potential bio-markers we used a set of 676 clinically relevant cancer functional events (CFEs[29]), across 17 different tissue types. The CFEs encompass mutations in cancer driver genes, amplifications/ deletions of chromosomal segments recurrently altered in cancer, hypermethylated gene promoters and microsatellite instability status. For each CFE and tissue type, we performed a Student's $t$-test for each selective gene dependency (SGD, Methods) contrasting two groups of cell lines based on the status of CFE under consideration (present/absent), for a total number of 2,142,162 biomarker/dependency pairs tested.

The total number of significant biomarker/dependency associations showed little variation across batch-correction methods at 5% FDR. However, a significantly larger number of biomarker/ dependency associations were identified when using CRISPR-cleanR compared to CCR-JACKS (largest $p$-value 1.0e-14, proportion test) or CERES (largest $p$-value 3.60e-10, proportion test) while little significant difference was found between CCR-JACKS and CERES (smallest $p$-value 0.038, proportion test) (Fig. 5a and Supplementary Data 3). Similar results were seen when the CFEs were split according to whether the biomarker was a mutation, recurrent copy number alteration or hyper-methylated region (Supplementary Fig. 3).

We next examined the ability of each dataset to recover known selective dependencies in individual cell lines. We downloaded a set of oncogenic gene alterations from OncoKB[30,31]. After filtering for genes that tend to be common essentials (mean dependency scores lower than $-0.5$ in the CRISPRcleanR-ComBat dataset, where $-1$ is the median of scores of known common essentials), we considered the oncogenes as positive controls in cell lines where they had indicated oncogenic or likely oncogenic gain of function alterations, and negative controls in all others. For each oncogene, we measured the NNMD between positive and negative cell lines (Fig. 5b). We found little difference in median performance by either pre-processing method or batch correction method. We then collected the dependency scores of all oncogenes in cell lines with a corresponding oncogenic

alteration and measured receiver-operator characteristic (ROC) AUC between them and the dependency scores of the same genes in cell lines without oncogenic alterations (Fig. 5c). By this measure, CRISPRcleanR outperformed CERES by 2.2% and CCR-JACKS by 4.0%, with minimal variations across batch correction method.

**Recovery of functional relationships.** We tested the ability of each dataset to identify expected dependency relations between paralogs, gene pairs coding for interacting proteins, or members of the same complex using gene pairs annotation from publicly available databases[32–34] (Methods). For each pair of genes known to have a functional relationship, we selected a random pair of genes with similar mean dependency scores across cell lines to serve as null examples. We calculated the false-discovery rate for the known pairs using the absolute Pearson correlation of their dependency profiles versus those of the null examples. Recovery of known relationships was unsurprisingly low, since many genes with known functional relationships do not exhibit selective viability phenotypes. ComBat + QN + PC1 or PC1-2 recovered the greatest number of expected gene-dependency relations at 10% FDR (Fig. 5d).

**Final selection of pre-processing methods and batch-correction pipelines.** Comparing the performance of batch correction methods across the use-cases we found that ComBat + QN out-performed ComBat alone and removing one or two principal components had similar or noticeable increases in performance compared to ComBat+QN. The principal component analysis indicated that ComBat+QN + PC1 corrected for linear and non-linear effects of technical confounders including assay length, guide library and media conditions. Removing the first two principal components offered little improvement over removing the first principal component alone and we found no attributable technical bias in the gene sets enriched in the second principal component. Overall, we selected ComBat + QN + PC1 as the batch correction pipeline as it had good performance overall metrics and a reduced impact on the data with respect to ComBat + QC + PC1-2, while still correcting for multiple technical biases. Comparing the pre-processing methods, we found that CERES outperformed the other methods while identifying essential genes

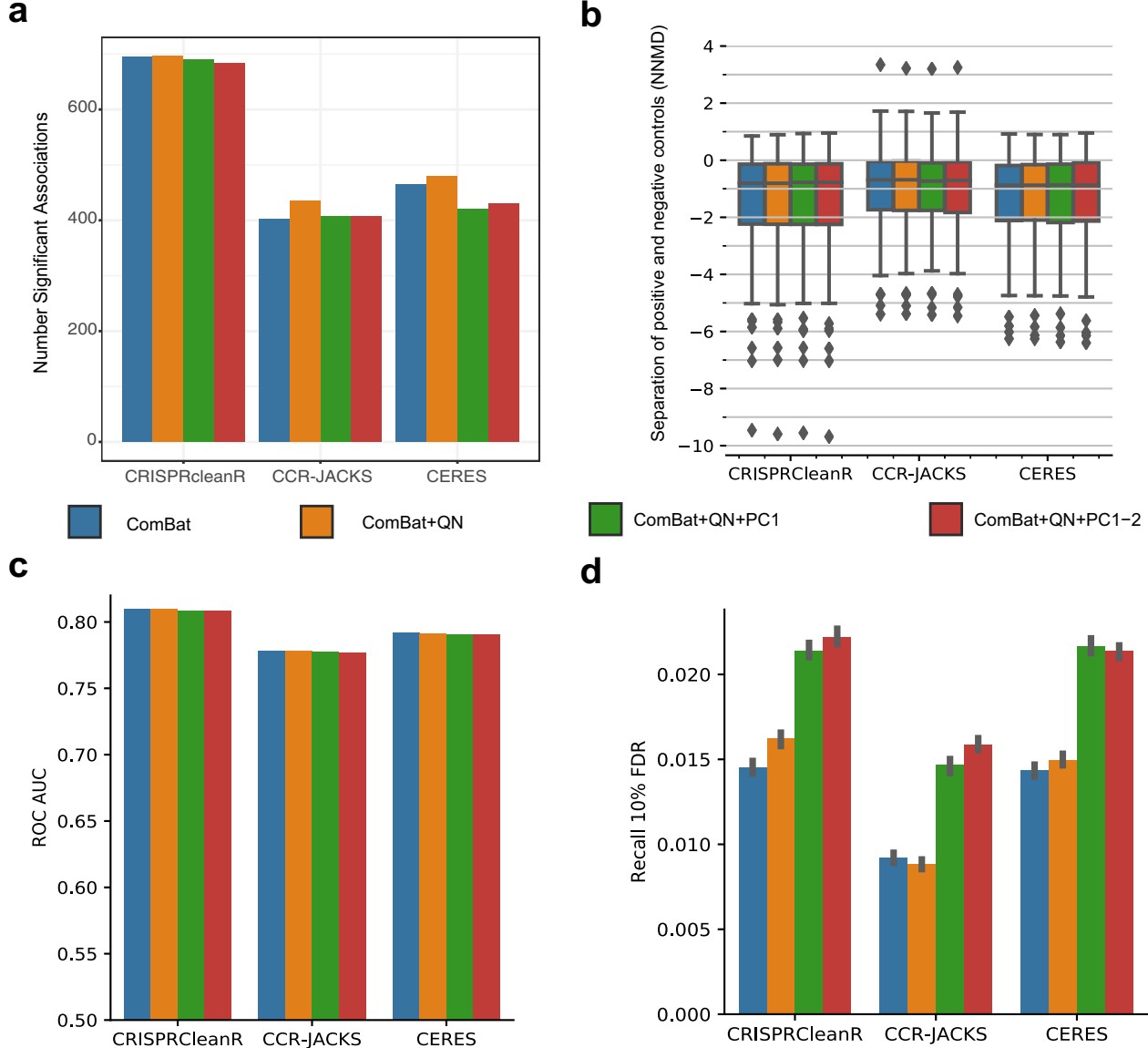

**Fig. 5 Use case Biomarkers and functional relationships. a** For each tissue pairs of Cancer Functional Events (CFEs) and dependencies were tested using a two-sided *t*-test for significant associations between the gene dependency and the absence/presence of a biomarker (CFE). The bar chart shows the total number of significant associations at 5% false-discovery rate (FDR) across tissue types for each of the integrated datasets. Results are shown across different pre-processing methods, CRISPRcleanR, CRISPRcleanR with JACKS (CCR-JACKS) and CERES (in different plots) and different batch correction pipelines. The different batch correction pipelines are ComBat only (ComBat in blue), ComBat followed by quantile normalization (ComBat+QN, orange), and ComBat+QN with either the first PC removed (ComBat+QN + PC1, shown in green) or first two PCs removed (ComBat+QN + PC1-2, red). **b** The per-oncogene null-normalized mean difference (NNMD) between cell lines with and without an indicated oncogenic gain-of-function indication (more negative is better). The boxplot shows the interquartile range in the box with the median as a horizontal line. Whiskers extend to 1.5 times the interquartile range and outlier points outside this range plotted. **c** For all identified oncogenes collectively, the receiver-operator characteristic (ROC) area under curve (AUC) between oncogene scores in cell lines where they have an indicated gain-of-function mutation and cell lines where they do not. **d**. For each dataset, the number of known gene-gene relationships recovered at 10% FDR. The number of relationships tested varied between $N = 346,114$ and $N = 346,744$. The error bars show the 95% confidence intervals generated from 100 bootstrap iterations.

and lineage subtypes, that CRISPRcleanR showed higher performance in the biomarker association use case, and these two methods performed comparably and better than CCR-JACKS in identifying known gene-gene relationships. In conclusion, we selected both CERES and CRISPRcleanR as processing methods and considered the two corresponding integrated datasets as the final results of our pipeline.

**Advantages of the integrated datasets over the individual ones**. In-line with the results from all the use-cases, we estimated the benefits of the integrated datasets with respect to the individual ones, in terms of increased capacity to unveil reliable sets of common essential genes (using CERES), as well as increased diversity of genetic dependencies and biomarker associations (using CRISPRcleanR).

To evaluate the increased coverage of molecular diversity and genetic dependencies in the integrated dataset we first estimated the increase in the number of detected gene dependencies with respect to the two individual datasets. To this aim, using the CRISPRcleanR processed dataset we quantified the number of genes significantly depleted in $n$ cell lines (at 5% FDR, Methods) for a fixed number of cell lines $n$ (with $n = 1, 3, 5$ or $n \geq 10$) of the integrated dataset, as well as in the individual Broad and Sanger datasets. The integrated dataset identified more dependencies, indicating greater coverage of molecular features and dependencies than in the individual datasets (Supplementary Fig. 4a).

We then evaluated the ability of the CERES processed integrated dataset to predict common essential genes and its performance when compared to the individual datasets and two existing sets of common essential genes from recent publications: Behan[2] and Hart[35]. We predicted common essential genes using two methods: the 90th percentile method[16] and the Adaptive Daisy Model (ADaM)[2]. The majority of genes called common essentials according to one of ADaM or 90th percentile methods was also identified by the other (1482 out of 2103, Supplementary Fig. 4b). We assigned to each of the 2,103 common essential genes a tier based on the amount of supporting evidence of their common essentiality. Tier 1, the highest confidence set comprised the 1482 genes found by both methods. Tier 2 had 621 genes found by only one method (Supplementary Data 4).

For each predicted set of common essential genes, we calculated recall rates of known essential genes sets obtained from KEGG[36] and Reactome[37] pathways. These pathways included Ribosomal protein genes, genes involved in DNA replication and components of the Spliceosome (Methods). The integrated set of common essentials (Tier 1 and 2) showed greater recall of known essential genes compared to Behan and Hart, and increased recall over the individual datasets for 5 out of the 6 gene sets (Fig. 6a).

We next generated a set of 647 genes that were never expressed across the panel of cell lines, to serve as high confidence negative controls (Methods). We then calculated the proportion of negative controls in each set of common essentials genes. The best performance was found with the Hart gene set (0%) followed by the integrated dataset (0.33%) (Fig. 6b). As the positive and negative controls did not cover all genes we further investigated the genes predicted to be common essentials. The integrated dataset predicted the largest number of common essentials, with 233 genes found in the integrated dataset alone. The 233 genes were enriched for Cell cycle genes (hypergeometric test, $q$-value 3.06e-9) and mitochondrial gene expression (hypergeometric test, $q$-value 3.66e-7), indicative of essential cellular processes. Similar results were observed for the 1,159 genes in the integrated set of common essentials but neither of the existing datasets (Behan and Hart) (Supplementary Table 1)

We next asked whether the CRISPRcleanR processed integrated dataset was able to unveil additional significant CFE/gene-dependency associations compared to either one of the Broad or Sanger (individual) datasets. Performing systematic biomarker analysis using CFEs on cell lines from individual tissue lineages unveiled 52 additional significant associations in the integrated dataset (when considering only CFE/gene-dependency pairs testable in the individual datasets at 1% FDR) with respect to those using the Sanger dataset alone, and 68 with respect to the Broad dataset (Supplementary Table 2). Examples included decreased dependency on *MDM2* in *TP53* mutant lung cell lines for the Sanger dataset, and increased dependency on *STAG1* in *STAG2* mutated central nervous system cancer cell lines for the Broad dataset (Fig. 6c). Furthermore, 19 tissue-specific significant associations identified in the integrated dataset were tested but not found significant in either the Broad or the Sanger dataset.

Two examples of these associations were increased dependency on *SMAD7* following loss of *CDKN2C* and *FAF1* in hematopoietic and lymphoid and increased dependency on *FOXA1* with gains of *CLTC* and *PPM1D* in breast (Fig. 6d).

**Sample size requirements for efficient data integration**. To further increase the coverage of a cancer dependency map, new CRISPR-cas9 screens should be integrated into the existing datasets as they are generated. To aid in this integration we estimated the minimum number of overlapping cell lines that should be screened to efficiently calculate and correct batch effects. We performed a downsampling analysis on the 168 cell lines screened at both Sanger and Broad, ranging from 5% to 90%, and used the obtained subset of cell lines to estimate and correct batch effects using ComBat. Following this, for each cell line DPG generated at either institute, we computed the Pearson correlation following batch correction using all 168 overlapping cell lines (Fig. 6e). We found a high degree of correlation between datasets at all levels of downsampling, with the minimum of eight samples still reducing batch effects when compared to no batch correction ($N = 0$) (Supplementary Fig. 4c). We next evaluated the batch correction using the average silhouette width (ASW) of the clustering induced by the institute of origin of the cell lines as a measure of the extent to which cell lines from the same institute clustered together. As expected, as the number of samples used to estimate and correct the batch effect decreases, the DPGs increasingly cluster by the batch of origin (Fig. 6f).

The ASW and Pearson correlation metrics both showed clear convergence with increasing sample size and at the same rate. Given the convergence of these metrics, the results showed that the 168 overlapping cell lines used were sufficient to maximize the batch correction using ComBat. Further the downsampling analysis showed convergence was reached at 90 cell lines and that between 30 and 40 cell lines would be sufficient to provide a batch-corrected dataset that is highly correlated (Pearson's Correlation over 0.995) with that obtained when estimating and correcting batch effects with using >90 cell lines.

The 168 overlapping cell lines were from 16 different lineages. To investigate the impact of lineage composition of the cell lines on the batch correction we also used a single lineage to estimate the batch effects. In the overlapping cell lines the lung lineage had the most cell lines (28 in total). We subsampled the lung cell lines to include 8, 17, or 25 cell lines (Supplementary Fig. 4d, e) and found little difference in performance between using a single and a mixture of lineages, indicating that this is not a major factor for estimating batch effects.

## Discussion

The integration of data from different high-throughput functional genomics screens is becoming increasingly important in oncology research to adequately represent the diversity of human cancers. Integrating CRISPR-Cas9 screens performed independently and/ or using distinct experimental protocols, requires correction and benchmarking strategies to account for technical biases, batch effects and differences in data-processing methods. Here, we proposed a strategy for the integration of CRISPR-Cas9 screens and evaluated methods accounting for biases within and between two dependency datasets generated at the Broad and Sanger institutes.

Our results show that established batch correction methods can be used to adjust for linear and non-linear study-specific biases. Our analyses and assessment yielded two final integrated datasets of cancer dependencies across 908 cell lines. In contrast to existing databases of CRISPR-Cas9 screens[38,39], our integrated

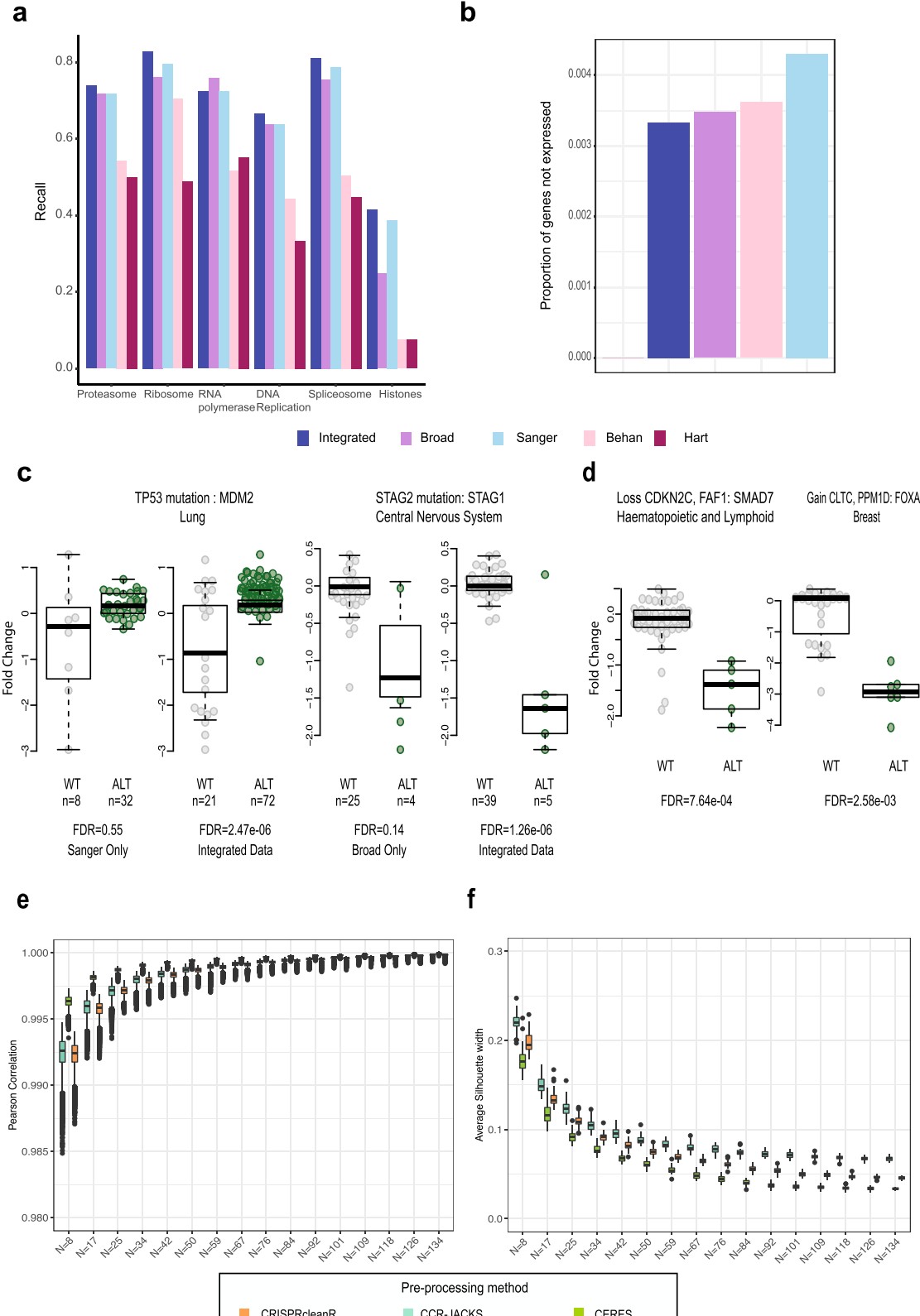

datasets are corrected for batch effects allowing for their joint analysis. Following integration, dependency profiles of cell lines from the same tissue lineage and cancer-specific subtypes show good concordance. Our integrated datasets cover a greater number of genetic dependencies, and the increased diversity of screened models allows additional associations between bio-markers and dependencies to be identified.

The integrated datasets were the output of two orthogonal pre-processing methods, CRISPRcleanR and CERES. The use-case analysis showed that CERES (which borrows information across screens) yields a final dataset better able to identify prior known essential and non-essential genes and clustering of cell lines by lineage. In contrast, CRISPRcleanR (a per sample method) was better able to detect associations between selective dependencies and

**Fig. 6 Advantages of an integrated dataset. a** Recall of essential genes sets for the integrated dataset (Integrated), across different tiers, compared to two previously published gene sets (Behan and Hart) and the two individual datasets (Broad and Sanger). **b** Proportion of genes in the common essential gene sets that are constitutively not expressed across the panel of cell lines and, therefore, likely to be false-positive results. **c** Examples of significant associations between genes and features, found in the integrated dataset compared to the individual dataset. We performed a two-sided *t*-test between each cancer functional event and gene dependency and multiple hypothesis corrected using Benjamini–Hochberg. The boxplot shows the interquartile range in the box with the median as a horizontal line, the whiskers extend to 1.5 times the interquartile range. The first boxplot shows an example of decreased dependency on *MDM2* in *TP53* mutant lung cell lines when comparing the Sanger dataset to the integrated dataset with $N = 8$ for Wild-type (WT) and $N = 32$ for altered (ALT) samples in the Sanger dataset and $N = 21$ (WT), $N = 72$ (ALT) in the integrated dataset. The second set of boxplots shows increased dependency on *STAG1* for *STAG2* mutated in central nervous system cancer cell lines found in the integrated dataset but not in the Broad dataset. The two-sided *t*-tests were performed with $N = 25$ (WT) $N = 4$ (ALT) in the Broad dataset and $N = 39$ (WT) $N = 5$ (ALT) in the integrated dataset. **d** Examples of significant associations found in the integrated dataset that were not significant in either of the individual datasets. The box contains the interquartile range with the median as a horizontal line. Whiskers extend to 1.5 times the interquartile range. In the integrated datasets there was an increased dependency on *SMAD7* in haematopoietic and lymphoid cancer cell lines with loss of *CDKN2C* and *FAF1* with $N = 46$ (WT) $N = 5$ (ALT) and $N = 28$ (WT) $N = 6$ (ALT) for increased *FOXA1* dependency associated with gain of *CLTC* and *PPM1D* in breast cancer cell lines. **e** The boxplots contain $N = 50$ random samples of between 5 and 90% of the 168 overlapping cell lines (number of cell lines in each sample indicated on the *x*-axis). For each sample the Pearson correlation of the DPGs following ComBat correction compared to the integrated dataset was calculated for each pre-processing method. **f** The average silhouette width (ASW) for each down sampled dataset ($N = 50$) was calculated using the institute of origin as the cluster label. An ASW of close to zero indicating a near random performance of the clustering, meaning the samples do not cluster by the origin of the screen and batch effects have been removed. The boxplot shows the interquartile range in the box with the median as a horizontal line. Whiskers extend to 1.5 times the interquartile range and outlier points outside this range plotted.

potential biomarkers, and had better recall of known oncogenic addictions. Therefore, results from both processing methods provide the best overall data-driven functional Cancer Dependency Map.

The data integration strategies and sample size guidelines outlined here can be used with future and additional CRISPR-Cas9 datasets to increase coverage of cancer dependencies. This will be important for oncological functional genomics, for the identification of novel cancer therapeutic targets, and for the definition of a global cancer dependency map. Further, as library design improves[24,40,41] we would expect the coverage and accuracy of the integrated datasets to also improve.

## Methods

**Pre-processing data**. Sanger data processed with CRISPRcleanR were obtained from the Score website (https://score.depmap.sanger.ac.uk/). The CRISPRcleanR corrected counts were used as input into JACKS, for the CCR-JACKS processing method. Raw counts and the copy number profiles for the Sanger dataset downloaded were processed with CERES[20]. The Broad data processed with CERES (unscaled gene effect) version 20Q2 scores were downloaded from the Broad DepMap portal[20]. The raw counts for Broad data 20Q2 were processed with CRISPRcleanR (v0.5) and the CRISPRcleanR corrected counts processed with JACKS. Gene names were matched across the Broad and Sanger datasets by updating both to the current version of HUGO gene symbols from the HGNC website. Missing entries were mean imputed for the principal component removal and then re-assigned as NA in the final matrix. Cell lines processed by both CERES and CRISPRcleanR were used for analysis. Tissue annotations for each cell line were obtained from the Cell Model Passports (https://cellmodelpassports.sanger.ac.uk/)[42].

**Batch correction pipelines**. The dependency profiles across genes (DPGs) for overlapping cell lines from each institute were first quantile normalized using the preprocessCore package (v1.48.0) in R[43]. Where an overlapping screen contained missing values, this screen was not considered in the batch correction estimation. Screen quality adjustments were made by fitting a spline (splines2 package v0.3.1 in R) to the average gene fold-change across cell line DPGs. Each DPG was then adjusted to remove the difference between the fitted spline and the diagonal. The overlapping cell lines were then batch-corrected using three different methods. A standard least squares model was fitted in R. The ComBat correction was performed using the sva package (v3.34.0) in R[44].

**Batch correction pipelines' assessment and weighted Pearson correlation metric**. Cell lines' rank neighborhoods were based on a weighted Pearson correlation metric. The weights were defined as the absolute mean (over the Broad and Sanger datasets) of a gene-dependency signal skewness across the 168 overlapping cell lines for the Broad and Sanger datasets. Using skewness upweights genes with a variable and sufficiently selective fitness profile while down weighting those that show weak/no-signal or unselective dependencies. Then for each query DPG we ranked all the others based on how similar they were to the fixed one in decreasing order, according to the wPearson scores. For each position $k$ in the resulting rank we then defined a $k$-neighborhood of the query DPG composed of all the other

DPGs whose rank position was $\leq k$. Finally, we determined the number of cell line DPGs that had the DPG derived from screening the same cell line in the other dataset (a matching DPG) in its $k$-neighborhood. The final rank for each cell line was defined based on the minimum rank obtained for each cell line when considering the DPG for that cell line from the Broad data compared to all DPGs, and similarly the DPG for the cell line in the Sanger dataset compared to all DPGs.

**Analysis of principal components**. The first two principal components (PCs) were extracted from ComBat corrected CRISPRcleanR data using the prcomp function in R from the stats package (v3.6.0). The top 500 genes (according to the absolute value of their PC loadings) were selected for enrichment analysis. The gene lists were used as input into the GSEA website (https://www.gsea-msigdb.org/) and were tested against the Gene ontology Biological Processes, Hallmark and Canonical Pathway databases. The top 10 significantly enriched (*q*-value < 0.05) gene sets were downloaded from the website.

**Batch correction extended to 908 cell lines**. The ComBat estimates, pooled mean, variance and empirical Bayes adjustments (mean and standard deviation) for each batch based on the analysis of 168 cell lines common to both initial datasets were computed. The ComBat correction using these estimates was then applied to all screens, i.e., the union of the two initial datasets. Particularly, each individual cell line DPG was shifted and scaled gene-wise using the batch correction vectors outputted by ComBat.

Further adjustments were then applied to all screens including quantile normalization, and the removal of either the 1st principal component of the joint datasets or the first two. Finally, DPGs for overlapping cell lines passing a similarity threshold (detailed below) were averaged. Across the three pre-processing methods the number of cell lines that matched their counterparts exactly after ComBat correction ranged from 51% to 86% (Fig. 3b), suggesting that under all pre-processing methods there remained cell lines whose DPGs diverged between studies. For each of the cell lines that matched their counterpart as the first neighbor we considered their distances (1-wPearson) as a measure of the variability in distance profiles between DPGs of the same cell line across institutes. We called divergent DPGs those with a distance greater than the 95th percentile of distances from matching cell lines. For 16 cell lines with divergent DPGs across all three processing methods, we selected the DPG from the screen with the highest quality to be included in the integrated datasets. As a quality metric we used the Null-normalized mean difference (NNMD, defined in the main text) and took its consensual value across the three datasets (resulting from applying CERES, CCR-JACKS and CRISPRcleanR).

**Agreement between dependency profile clusterings and cell line tissue labels**. We selected 500 genes with the highest variance in the CERES ComBat integrated dataset and performed repeated 100 $k$-means clusterings cell lines using the high variance genes for each pre-processing and batch-correction method. For each clustering, we calculated the adjusted mutual information between the obtained clusters and the cell line tissue labels as specified in the annotation provided by the sample_info file of the DepMap_public_20Q2 dataset[20] using sklearn's (v0.23.1) python function adjusted_mutual_info_score (https://scikit-learn.org/stable/).

**Recall of known gene relationships**. We assembled a set of functionally related gene pairs using paralogs identified by EnsemblCompara[32], protein-protein interactions identified by Li et al.[33], and CORUM complex comemberships[34]. For a given dataset, for each pair of related genes, we calculated a Pearson correlation coefficient between those genes' dependency scores across cell lines. We then binned each gene that appeared in the list of known gene relationships according to its mean gene score using 20 equally spaced bins. For pairs of genes in the related genes pairs, we chose one as the query gene and replaced its related partner with another randomly selected gene of similar gene mean, i.e., belonging to the same bin, excluding genes known to be related to the query gene. We calculated Pearson's correlation coefficients between these randomly selected gene pairs to generate a null distribution, from which we calculated empirical p-values and Benjamini–Hochberg FDRs for known related gene pairs. Ensuring that the pairs of genes used in the null distribution have similar distributions of mean gene effect as the pairs of known related genes is necessary because variable screen quality can produce a high but artefactual correlation between any pair of common essential genes, and CORUM is highly biased towards common essentials. This is discussed further in the comparisons of batch corrections in Dempster et al.[28].

**Unexpressed false positives**. We defined a gene as unexpressed in a cell line if the $log_2$(Transcripts per million +1) of its DepMap expression was <0.01[45]. Any score of an unexpressed gene in a cell line was called a false positive if it fell in the bottom 15% of gene scores for that cell line.

**Identifying selective dependencies**. NormLRT and likelihood of normal distribution was calculated in R using the MASS package[46] (v7.3-51.4). For the skew t-distribution the st.mple function from the sn package (v1.6-3) was used to calculate the likelihood. If the fitting procedure failed different degrees of freedom were used iteratively until a solution was found. The degrees of freedom used in order were 2, 5, 10, 25, 50, and 100.

**Systematic association test between molecular features and gene dependencies**. We performed a systematic two-sided unpaired Student's t-test (with the assumption of equal variance between compared populations) to assess the differential essentiality of each gene across a dichotomy of cell lines defined by the status (present/absent) of each CFE in turn. We tested genes whose NormLRT values were greater than 200 in any integrated dataset. From these tests, we obtained p-values against the null hypothesis that the two compared populations had an equal mean, with the alternative hypothesis indicating an association between the tested CFE/gene-dependency pair. p-values were corrected for multiple hypothesis testing using Benjamini–Hochberg (method "fdr" using the p.adjust function in R (v3.6.0)). We also estimated the effect size of each tested association using Cohen's Delta (ΔFC), i.e., the difference in population means divided by their pooled standard deviations.

**Evaluating known selective dependencies**. A table of all annotated oncogene variants was downloaded from OncoKB[31] (http://oncokb.org/api/v1/utils/allAnnotatedVariants). The table was filtered first for genes that were (likely) oncogenic and alterations that were (likely) gain-of-function or switch-of-function. For each alteration, the DepMap public 20Q2[20] mutation and fusion calls were used to identify which cell lines had the alteration. These cell lines were treated as positive controls for the gene in question, with all other cell lines treated as negative controls. Only oncogenes with at least one positive cell line were retained. For each integrated dataset, we calculated the ROC AUC between all positive oncogene-cell line pairs and negative pairs. Then, for each oncogene with at least two positive cell lines, we calculated the NNMD between its positive and negative cell lines.

**Identification of common essential genes via the 90th percentile method**. The 90th percentile method[16] finds for each gene the cell line on the boundary of its 90th percentile least dependent cell lines. It then calculates the rank of that gene in that cell line, by sorting all the genes based on their dependency score in increasing order. A mixture of two normal distributions is then fitted to the rank positions of all genes. Those genes with ranks below the crossover point of these two distributions are labeled as common essentials.

**ADaM method**. Binary depletion matrices for the integrated datasets were calculated as outlined in the next section and used with the ADaM method (ADaM2 R package v0.1.0) as described in Behan et al.[2]. The ADaM method determines the number of cell lines dependent on a gene required to call that gene common essential. The number of cell lines is calculated by maximizing the tradeoff between true-positive rate (using a set of known prior essential genes) and the deviance from the null expected rate (calculated using random permutations of the binary depletion matrix). Common essential genes were identified for each tissue separately (according to the cell line annotation from the Cell Model Passports[42]) and were then used as input into ADaM to determine pan-cancer common essential genes.

**Binary depletion calls**. Binary depletion calls were computed by considering each cell line DPG as a rank-based classifier of essential/non-essential genes[11] (with gene rank positions determined by their fitness effect, i.e., average depletion fold-change of targeting single-guide RNAs abundance at the end of the assay with respect to plasmid counts).

The fitness effect threshold was then fixed as that corresponding to the largest rank position $r$ guaranteeing a false-discovery rate (FDR) < 5%, when the predicted essential genes are those with a rank position $\leq r$. This allowed us to assign to each gene in each cell line, in each of the two datasets, a binary dependency score.

To identify significantly depleted genes for a given cell line at a 5% FDR, we ranked all the genes in the cell line DPG in increasing order based on their depletion log fold-changes. We used the ranked list to calculate the precision curve using a set of prior known essential ($E$) and non-essential ($N$) genes, respectively, derived from Hart et al.[11].

To estimate the rank position corresponding to the 5% FDR threshold we calculated for each rank position $k$, a set of predicted essential genes $P(k) = \{s \in E \cup N: r(s) \leq k\}$, with $r(s)$ indicating the rank position of $s$, and the corresponding positive predictive value (or precision) PPV$(k)$ as:

$$PPV(k) = |P(k) \cap E|/|P(k)| \qquad (1)$$

We then determined the largest rank position $k^*$ with PPV$(k^*) \geq 0.95$ (equivalent to a FDR $\leq 0.05$). The 5% FDR logFCs threshold $F^*$ was defined as the logFCs of the genes such that $r(s) = k^*$. We called all genes with a logFC $< F^*$ as significantly depleted at 5% FDR.

Binary dependency matrices were defined as gene by cell lines matrices with non-null entries corresponding to significant dependency genes at 5% FDR, for each cell line, i.e., column.

**Positive controls for common essentials**. To generate sets of prior known common essential genes we downloaded gene sets from MsigDB (v7.2) using the R package qusage (v2.20.0). The gene sets used were from KEGG were KEGG_SPLICEOSOME, KEGG_RIBOSOME, KEGG_PROTEASOME, KEGG_RNA_POLYMERASE and KEGG_DNA_REPLICATION. For the histone gene set we combined two Reactome gene sets REACTOME_HATS_ACETYLATE_HISTONES and REACTOME_HDACS_DEACETYLATE_HISTONES as well as the curated histones gene set from[2].

**Negative controls for common essentials**. We compiled a set of negative controls for the common essential genes as those genes that were not expressed across all cell lines. We defined a gene as unexpressed across the panel of cell lines using the $log_2$(Transcripts per million +1) of its CCLE expression[20] and the 90th percentile method (The input into the ADaM2 package (available at https://github.com/DepMap-Analytics/ADAM2) performing the 90th percentile method was $-1 * log_2$(Transcripts per million+1) to ensure correct ranking). A gene defined as constitutively unexpressed was, therefore, one that was still lowly expressed in its highly ranked (90th percentile) most expressed cell line.

**Downsampling for batch correction sample sizes**. We down sampled 50 times the overlapping cell lines at different levels between 5 and 90%. Random samples were generated using probabilities of selecting a cell line based on the relative proportions of each cell line lineage in the overlapping dataset. Using the down sampled set of overlapping cell lines ComBat was used to calculate the batch adjustment vectors. The batch adjustment vectors were then applied to all 1074 cell lines. The correlation of a cell lines fold changes batch-corrected using the down sampled datasets and the full 168 overlapping cell lines was calculated and compared to the correlation with no batch correction.

To evaluate the batch correction, we also used the average silhouette width as a measure of clustering. We calculated the average silhouette width for each batch-corrected dataset (using samples of the overlapping cell lines) using the institute of origin as the cluster label. The average silhouette width is 1 for perfect clustering (or complete separation of cell lines by the institute of origin) with 0 indicating random performance of the clusters.

## Data availability

The final integrated datasets are available in Figshare with the data identifier https://doi.org/10.6084/m9.figshare.c.5289226.v1 (CRISPR combined dataset) and at https://depmap.org/broad-sanger/. Integrated datasets are also accessible through the DepMap (https://depmap.org) and Score (https://score.depmap.sanger.ac.uk) web-portals. Data from OncoKB is accessible through http://oncokb.org/api/v1/utils/allAnnotatedVariants.

## Code availability

Scripts and software packages implementing the integration pipeline described in this manuscript and needed to reproduce results and figures are available on GitHub at https://github.com/DepMap-Analytics/IntegratedCRISPR code identifier https://doi.org/10.5281/zenodo.4497774 with data sources available on Figshare: https://figshare.com/projects/Integrated_CRISPR/78252 with data identifier https://doi.org/10.6084/m9.figshare.c.5289226.v1.

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

## Acknowledgements

This work was partially funded by Open Targets [project OTAR0255] and by the Wellcome Trust [grant 206194]. We thank Leo Parts for a number of insightful discussions.

## Author contributions

C.P. conceived the study, designed, implemented and performed analyses, assembled figures, curated data, wrote the manuscript. J.M.D. conceived the study, designed, implemented and performed analyses, assembled figures, and contributed to manuscript writing. I.B. contributed to pipeline implementation. E.G. performed analyses, assembled figures, revised the manuscript. H.N. assembled figures, revised the manuscript. E.K., D.v.d.M., A.B., H.L., P.J. contributed to data curation. J.M.M., M.J.G., and A.T. revised the manuscript and contributed to study supervision. F.I. conceived the study, designed analyses, contributed to figure production, wrote the manuscript, acquired funds, and supervised the study.

## Competing interests

M.J.G. and F.I. receive funding from Open Targets, a public-private initiative involving academia and industry. M.J.G. receives funding from AstraZeneca and performs consultancy for Sanofi. F.I. performs consultancy for the joint CRUK—AstraZeneca Functional Genomics Center. A.T. is a consultant for Tango Therapeutics and Cedilla Therapeutics. J.M.D., J.M., and A.T. receive funding from the Cancer Dependency Map Consortium, but no consortium member was involved in or influenced this study. All the other authors declare no competing interests.
