## [Peer Review File · Nature Communications]

REVIEWERS' COMMENTS

Reviewer #1 (Remarks to the Author):

The revised manuscript has been significantly improved both in structure, statistical analysis and representation, examples and recommendations. The availability of the datasets, integration with DepMap and Score and the code availability now make this work of general interest and use. The inclusion of a more up-to-date and a larger data-set makes the comparisons with the integrated dataset even more compelling.

The important comment of reviewers was the lack of conclusive evidence for choosing one over the other pre-processing pipeline. With the inclusion of use cases on different classes of dependencies, the authors provide convincing evidence for their different abilities to identify dependencies. It is indeed up to the user to select the most appropriate for their specific question. This aspect is properly addressed and presented in the current manuscript.

The extension of the analysis of the essential and not essential genes provides a stronger basis for the conclusions based on the comparisons of the different methods and dat-sets.

The inclusion of the the analysis of subsampling is an important aspect, specifically for the incorporation of novel, to be generated datasets.

Reviewer #3 (Remarks to the Author):

I have re-read the comments and manuscript and I am satisfied that the authors have addressed my concerns. I think this work presents a good contribution and will hopefully serve as a useful resource for the community. I hope that, as the authors said, the data will in fact be made queryable through their existing portals.